# “Singing” Multilayer Ceramic Capacitors and Mitigation Methods—A Review

**DOI:** 10.3390/s22103869

**Published:** 2022-05-19

**Authors:** Corina Covaci, Aurel Gontean

**Affiliations:** Applied Electronics Department, Politehnica University Timisoara, 300006 Timișoara, Romania; etc@upt.ro

**Keywords:** singing capacitors, MLCC, acoustic noise, electronics design, PCB acoustics measurement methods, simulation, analysis, IoT sensors

## Abstract

Multilayer Ceramic Capacitors (MLCC) have a major role in modern electronic devices due to their small price and size, large range of capacitance, small ESL and ESR, and good frequency response. Unfortunately, the main dielectric material used for MLCCs, Barium Titanate, makes the capacitors vibrate due to the piezoelectric and electrostrictive effects. This vibration is transferred to the PCB, making it resonate in the audible range of 20 Hz–20 kHz, and in this way the singing capacitors phenomenon occurs. This phenomenon is usually measured with a microphone, to measure the sound pressure level, or with a Laser Doppler Vibrometer (LDV), to measure the vibration. Besides this, other methods are mentioned in the literature, for example, the optical fiber and the active excitation method. There are several solutions to attenuate or even eliminate the acoustic noise caused by MLCC. Specially designed capacitors for low acoustic levels and different layout geometries are only two options found in the literature. To prevent the singing capacitor phenomenon, different simulations can be performed, the harmonic analysis being the most popular technique. This paper is an up-to-date review of the acoustic noise caused by MLCCs in electronic devices, containing measurements methodologies, solutions, and simulation methods.

## 1. Introduction

Capacitors are passive electronic components found in various shapes and using different materials [1]. Numerous ceramic capacitors, especially multilayer ceramic capacitors (MLCC), are used on a modern printed circuit board (PCB). They have a major role in resonant circuits, power supply bypass, and filters [2]. This makes them indispensable in all modern electronic devices, including but not limited to wearable smart sensors, environmental monitoring, agriculture, and food control, to name a few. Due to their popularity, the global market of MLCCs was valued at USD 5315 million in 2017 and it is predicted to increase to USD 7833 million by 2024 [3,4].

As presented in Figure 1, MLCCs are composed of three main elements: inner electrode, outer electrode, and ceramic dielectric material, mainly made of nickel, silver, and palladium [5].

The main advantages of the MLCCs are their small size and price [6]. Despite their small size, they have large capacitance and favorable electrical characteristics. These characteristics are small equivalent series inductance (ESL), small equivalent series resistance (ESR), good frequency response, a wide range of capacitance value, and their ability to be used for long periods at high temperature or in high-voltage applications [7,8,9].

All of these MLCCs advantages are due to the high-level permittivity dielectric material, Barium Titanate (BaTiO_3_), they are made of [10,11,12]. Ironically, the two main electro-mechanical properties of the BaTiO_3_ cause one of the newest problems in electronic devices: the singing capacitors phenomenon [13]. These properties are piezoelectricity and electrostriction. When an AC voltage is applied to the MLCC, the capacitor starts to vibrate due to the piezoelectricity. At the same time, the electric field generated between the inner electrodes creates electrostrictive vibration whose level is similar to the piezoelectric vibration level. Furthermore, the nonlinear phenomenon of the electrostriction also makes a second harmonic frequency vibration of the applied voltage [13]. In recent years, the thickness of the hundreds of dielectric layers present in an MLCC has decreased to achieve a large capacitance in a small package size [1]. Therefore, both properties must be taken into consideration, as the vibration generated by the piezoelectric effect is proportional to the electric field, and the vibration generated by the electrostriction is proportional to the square of the electric field [8].

Due to the thin dielectric layer, the *x* and *y* direction components of the electric field applied to an MLCC can be ignored. Therefore, in the equation of mechanical strain Equation (1), only the *z*-direction is taken into consideration [8].
(1)$$s_{z}=d_{33}\cdot E_{z}+M_{33}\cdot E_{z}^{2}$$
where *s_z_* is the mechanical strain, *d_33_* is the piezoelectric coefficient, *M_33_* is the electrostrictive coefficient, and *E_z_* is the applied electric field on the *z*-direction (the electric field applied on the *x* and *y* directions, *E_x_* and *E_y_*, respectively, can be neglected) [1,8,13,14].

In general, the electric signal applied on MLCC has DC and AC components, as shown in Equation (2). In this case, the mechanical strain equation is expressed using Equations (3) and (4) [1,8,13,14].
(2)$$E_{z}=E_{DC}+E_{AC}\cdot cos\omega t,$$
(3)$$s_{z}=d_{33}\cdot \left(E_{DC}+E_{AC}\cdot cos\omega t\right)+M_{33}\cdot {\left(E_{DC}+E_{AC}\cdot cos\omega t\right)}^{2},$$
(4)$$s_{z}=\left(d_{33}\cdot E_{DC}+M_{33}\cdot E_{DC}^{2}+\frac{1}{2}\cdot M_{33}\cdot E_{AC}^{2}\right)+\left(d_{33}+2\cdot M_{33}\cdot E_{DC}\right)\cdot E_{AC}\cdot cos\omega +\frac{1}{2}\cdot M_{33}\cdot E_{AC}^{2}\cdot cos2\omega t.$$

As shown in Equation (4), although the applied electric field only has a single frequency term, we can observe the deformation at the second harmonic frequency due to the electrostriction. When the DC component is zero (the electrical signal contains only the AC component), the vibration is caused only by piezoelectricity. As the vibration at the fundamental frequency is influenced both by piezoelectricity and electrostriction, the nonlinear characteristic of the MLCC is present only when the DC electric field is not zero [8,14].

The vibration level of the MLCCs depends on the number of the inner layers, the applied voltage, and the piezoelectric coefficient of the dielectric material [10,15]. If we check the capacitance *C* Formula (5), we can observe that it depends on the number of layers [15]. Therefore, we can declare that the vibration of the MLCC is proportional to the capacitance [7].
(5)$$C=\varepsilon_{0}\cdot \varepsilon_{r}\cdot \frac{S}{d}\cdot N,$$
where *C* is the capacitance, *ε_0_* is the permittivity in a vacuum, *ε_r_* is the relative permittivity of the dielectric, *S* is the electrode area, *d* is the dielectric layer thickness, and *N* is the number of layers [15].

Besides the piezoelectric and electrostrictive effects, the converse magnetoelectric effect also influences the MLCCs behavior. An induced electric polarization that appears under an applied magnetic field characterizes the magnetoelectric effect. On the contrary, the converse magnetoelectric effect is characterized by an induced magnetization under an external electric field [16]. When driven near resonance frequency, the converse magnetoelectric coefficient reaches the maximum [17]. The resonance frequency of the MLCC vibration is in the range of MHz, therefore we should not be able to hear anything. However, as the MLCCs are surface-mount devices (SMD), the induced vibration is transferred to the PCB via solder joint [1,18]. When an AC voltage is applied to the MLCC, the dielectric material expands in the direction of the electric field and contracts in the direction perpendicular to the electric field, causing the deformation of the board, as shown in Figure 2 [4]. Therefore, the PCB starts to vibrate with the MLCC, and the frequency can reach the audible range of 20 Hz–20 kHz [10].

We can conclude that the singing capacitor phenomenon is caused by three major factors [2]:the MLCC itself—the capacitor acts as an excitation source;the mounting situation—the solder joint is the vibration transfer path;the PCB—the board is the acoustic noise resonator.

There are several solutions to attenuate or eliminate the acoustic noise caused by MLCCs. Figure 3 highlights the singing capacitor phenomenon and a summary of the possible mitigation solutions. The paper is structured around both aspects.

Some studies suggest that superior dielectric properties are achieved at higher temperatures [19,20]. However, the BaTiO_3_ material does not present significant variations up to the Curie temperature, around 130 °C [21,22]. In these conditions, we can state that in the normal operation of electronic devices, the acoustic noise caused by MLCCs is not influenced by temperature.

In this paper, a review of the literature information about the singing capacitors phenomenon is presented. In Section 2, we will show how to detect the problematic MLCCs on a PCB. In Section 3, the solutions for singing capacitors found in the literature will be presented, while in Section 4, we will present types of simulation and analysis to prevent audible noise on PCB. We will end with a short discussion about the paper’s highlights and findings.

## 2. Measurement Methodologies

There are two main ways to measure the singing MLCCs: measure the acoustic noise or the vibration. The best way to characterize acoustic noise is by measuring the sound pressure level (SPL). Usually, this investigation needs a microphone probe, an FFT analyzer, and an anechoic box/room [4]. The microphone has two sensors: one pressure sensor to measure the sound pressure in the air and one velocity sensor to measure the velocity of the motion of the air [9]. The FFT analyzer is used to obtain the SPL spectrum, and the anechoic box is used to reduce the external acoustic noise that might influence the measurement. The SPL is defined as presented in Equation (6) [4,7]:(6)$$SPL=20\cdot \log\left(\frac{P_{RMS}}{P_{0}}\right),$$
where *P_RMS_* is the root mean square (RMS) deviation from the background atmospheric pressure and *P_0_* is the reference level.

Unfortunately, we cannot identify the problematic capacitors using SPL measurement. For detecting the capacitors which trigger the PCB vibration and produce the acoustic noise, the most suitable solution is scanning the PCB using a Laser Doppler Vibrometer (LDV). It uses the detection Doppler shift of the reflected light to measure the vibration of a surface without contact [4,9]. The LDV has a precise resolution of the submillimeter range, making it suitable for small-size MLCCs measurements.

Measuring the vibration is more convenient and easier to implement than measuring the acoustic noise. Therefore, we can measure the vibration using LDV and after that correlate the results with acoustic noise.

Ko et al. [7] calculated the correlation criteria between acoustic noise and the vibration of the PCB, by measuring the *SPL* and *V_cb_* on 30 samples. The result was a linear relation represented in Equation (7):(7)$$SPL=0.57\cdot V_{cb}+127.30$$

In conclusion, the acoustic noise can be predicted by the vibration response of the PCB. We also know that the electrical signal through the MLCC causes vibration due to the electromechanical characteristics of the BaTiO_3_. Therefore, it is important to correlate the acoustic noise with the electrical signal.

Another method to measure vibration is an optical fiber sensor. This method measures vibration up to tens of kHz by using glass fiber bundles to illuminate the target and collect the reflected light [23,24], and it is considered suitable due to the optical fiber sensor technology’s high measurement accuracy and lack of electromagnetic interference [25]. Although this method could be used to measure the PCB vibration caused by the MLCCs, the amount of information available in the literature is insufficient. Moreover, the existing papers which study the optical sensor fiber have outdated information. For example, Perrone and Vallan [23] suggest in their article written in 2009 that LDV is not precise enough to measure very small displacements, while Sun et al. [9,10] demonstrate the opposite in their papers written in 2019 and 2020.

Of course, another classical method to measure vibration is a piezoelectric accelerometer [26]. Unfortunately, this cheap and popular sensor can influence the measurement due to its size and weight [23].

Sun et al. [10] proposed two methods to correlate the electrical signal with the acoustic noise while identifying the problematic MLCCs: Active Excitation Method and Vibration and Rail Voltage Coherence Method. The former method detects the problematic MLCCs at specific frequencies with a high level of signal-to-noise ratio. The latter method tracks the transient event on the signal while the system is working in normal operation mode.

During the Activation Excitation Method, the device under test (DUT) is turned OFF, while a signal generator externally excites the MLCCs via an audio power amplifier. The same signal generator is also connected to the vibration control module of an LVD to eliminate the irrelevant vibration signal. The laser head scans each MLCC at different frequencies and the LDV creates a vibration color map. Using this method, we can identify the problematic MLCCs at specific frequencies by observing the MLCC locations with higher vibration strength [10].

Unlike the Activation Excitation Method, the Vibration and Rail Voltage Coherence Method analyses the DUT under normal operation mode to track the effect of the electrical signal transient on the MLCCs. When using this method, the signal’s noise voltage and the MLCC vibration response are examined by a coherence function defined as presented in Equation (8):(8)$$C_{xy}\left(f\right)=\frac{{\left|G_{xy}\left(f\right)\right|}^{2}}{G_{xx}\left(f\right)\cdot G_{yy}\left(f\right)},$$
where *C_xy_(f)* is the coherence between the voltage signal *x(t)* and the vibration signal *y(t)*, *G_xy_(f)* is the cross-spectral density between *x* and *y*, and *G_xx_(f)* and *G_yy_(f)* are the auto spectral density of *x* and *y*, respectively [10].

The coherence value is always subunit, therefore an ideal linear system with *x* as a single input and *y* as a single output will be 1, while for a system with *x* and *y* completely uncorrelated the value will be 0.

The measurement setup for the Vibration and Rail Voltage Coherence consists of a passive high impedance voltage probe connected to the LDV’s laser controller. The voltage probe accesses the electrical signal through a pair of thin metal wires, so as to not influence the board vibration, and a laser head points to the center of each MLCC.

For each MLCC, the voltage signal spectrum is captured, and the corresponding vibration response is plotted; therefore we can calculate the coherence value at different frequencies. When the coherence value is close to 1, it means that the produced vibration is generated mostly by the electrical signal. When the coherence value is close to 0, it means either that the captured output *y* is mostly noise, or it has no phase consistency with the input *x* [10].

In the next chapter, we present several solutions to reduce or eliminate the singing capacitors effect.

## 3. Solutions

Several solutions are presented in the literature for the singing capacitors phenomenon, such as alternative MLCC types, different orientation and position geometry on PCB, solder joint reduction, and others. Moreover, significant capacitor suppliers are aware of this issue and present methods to attenuate or eliminate the acoustic noise caused by MLCCs. In this chapter, we will go through the solutions currently available.

### 3.1. MLCC Manufactures Solutions

Murata™ comes with three capacitor series to suppress the singing capacitor phenomenon [27]:KRM series—uses metal terminals to raise the capacitor above the PCB (Figure 4a,b);ZRA/ZRB series—the capacitor is mounted on an interposer substrate that absorbs the vibration before reaching the PCB (Figure 5);GJ4 series—uses materials with a lower dielectric constant to attenuate the vibration between the inner layers.

Texas Instruments™ also recommends Murata™’s low acoustic noise MLCCs series. According to Texas Instruments™, by using the low distortion dielectric, acoustic noise is reduced by 7 dB; when using capacitors with an interposer, we eliminate 13 dB; and by using metal terminal capacitors, the noise caused by the MLCC is reduced by 25 dB [28].

Although Murata™’s capacitors are effective, Texas Instruments™ warns users about their high price. Ko et al. [2] not only agree with Texas Instruments’ statement, but also say that using metal terminal capacitors requires additional manufacturing processing steps. Otherwise, the MLCCs may fall off the PCB due to an insufficient amount of solder paste.

Besides using MLCCs designed to suppress acoustic noise, Texas Instruments™ also recommend some design changes, such as shifting the vibration frequency by using a thicker PCB, placing the components at the edge of the PCB, placing the capacitors symmetrically on top and bottom, or improving the load-transient response or line-transient response [28].

Samsung™ also developed three special MLCC series to reduce the singing capacitors phenomenon [29]:THMC series—has a thicker dielectric layer at the bottom of a typical MLCC (Figure 6);ANSC-A series—uses an alumina substrate between the MLCC and PCB (Figure 7);ANSC-B series—has metal plates attached to the capacitor’s terminals (Figure 8).

The ANSC-A and ANSC-B series are more efficient than the THMC series due to the separate structure, which isolates the vibration better than the internal dielectric layer. However, the THMC series is more suitable for applications with height limitations [29].

TDK™, on the other hand, offers a halogen-free series with dipped radial leads capacitors. This series consists of resin-coated ceramic capacitors with two lead wires soldered on the terminals, as shown in Figure 9. These capacitors comply with AEC-Q200 automotive standards, are an effective countermeasure against acoustic noise, and alleviate the mechanical and thermal stress on the MLCC [30]. However, the size of the dipped radial leads capacitors and the fact that they are through-hole capacitors makes them unsuitable for designs with dimensions restrictions.

Although these capacitors could be a solution to resolve the singing capacitors phenomenon issue, the price impact is significant, especially if we consider that multiple MLCCs cause the acoustic noise.

### 3.2. Layout Optimization

There are several layout configurations for MLCCs. Sun et al. [31] studied three different layout geometries:L-shape and T-shape configuration—capacitors are placed perpendicular to each other (Figure 10);Parallel configuration—the MLCCs are parallel, placed on the same side of the PCB (Figure 11);Mirror or Back-to-Back configuration—capacitors are on opposite sides of the PCB (Figure 12).

The authors of Ref. [31] investigated these geometries using four capacitor types:Regular capacitor—standard MLCC shape;3-terminal capacitor—MLCC with an additional terminal, placed between the classical soldering pads;Reverse geometry capacitor—MLCC with terminals placed on the longer edge;Interposer capacitor—MLCC described in the previous subchapter.

Due to the piezoelectric effect, the MLCC will contract and expand when a voltage is applied. The vibrations caused by two capacitors can be cancelled depending on their relative location and the in-phase or out of phase power ground pins arrangements. When placed in an L-shape or T-shape layout configuration, the vibrations of the two MLCCs are orthogonal to each other and provide a certain level of cancellation. In the parallel layout geometry case, the out of phase power-ground pins configuration allows the PCB vibration cancellation. Due to the top–bottom PCB symmetry, the mirror or back-to-back configuration can cancel the vibration when the voltages are in-phase. During their experiment, Sun et al. [31] applied two sine electrical signals in or out of phase to the MLCCs, and the acoustic noise was measured.

The first observation was that, in comparison with regular capacitor type, the three special types of capacitors reduced the level of acoustic noise up to 10 dB [31]. The reverse geometry type had the smallest impact, while the 3-terminal capacitor and the interposer types had the more significant reduction in noise.

The second observation was that the layout geometry did not have the same effect for all the capacitor types. The results are presented in Table 1 [31]:

Due to the cost implications, the back-to-back configuration is most appreciated in the literature. We can say that, for most applications, the singing capacitors phenomenon is reduced by using the mirror layout geometry and cheap, classical MLCC.

### 3.3. Other Methodes to Attenute the Singing Capacitors Phenomenon

As stated in the introduction chapter, the solder joint is the vibration transfer path from the MLCC to the PCB. Therefore, we can assume that a smaller solder paste quantity means less acoustic noise. Unfortunately, Sun et al. [32] demonstrate the opposite of this. The authors used three stencils with different heights: 3 mils, 4 mils, and 5 mils. No significant improvement was observed.

Another solution frequently found in the literature is placing the capacitor in a vertical orientation on PCB. When the inter electrode plates are perpendicular to the PCB, MLCC is considered to be placed in the vertical orientation. Otherwise, when the plates are parallel to the PCB, the capacitor is considered to be placed in horizontal orientation. This theory results from the fact that the MLCC placed in horizontal orientation behaves as an open-circuited transmission line, driven from its opposite end. The vertically oriented capacitors behave as an open-transmission line open-circuited at each end, driven from the center, and thus the resonating frequency is higher than in the horizontally oriented case [33].

Ko et al. [2] propose a solution to attenuate the acoustic noise by increasing the MLCCs’ cover layer. This process does not change the electrical properties of the capacitor, but reduces the vibration transmitted to the PCB. However, the authors warn that the suitable cover layer thickness differs from one application to another. In their experiment, the cover layer had to be thickened by 10% to observe a significant improvement.

This chapter presented different solutions to the singing capacitors issue. In the next chapter, we will present several simulation and analysis methods to prevent the apparition of the acoustic noise caused by MLCCs.

## 4. Prevention

Usually, in electronics, it is easier to prevent than to correct. Besides the effort and time needed to find the root cause and resolve it, sometimes it is impossible to make any layout changes. Therefore, it is recommended to simulate the design before implementing it.

The most frequently found simulation in literature is the modal analysis. However, this analysis only simulates the intrinsic vibration characteristic of the PCB without taking into consideration the influence of the vibrating MLCCs [34]. Therefore, for a good understanding of the singing capacitors phenomenon, the model should include three pieces of information: layers and material properties of the board design to evaluate the impact on the stiffness, the PCB fixation into the product design, and components information [4]. In this case, we need a harmonic analysis instead of the standard modal analysis.

Sun et al. [34] presented a design guideline for harmonic analysis to reduce the acoustic noise caused by MLCCs. Instead of using a lumped uniform mass to model the dielectric material in the multilayer PCB, they used the trace mapping technique to simulate the copper trace pattern without exaggerating the model complexity. After they performed the modal analysis to identify sensitive spots of the board to the external vibration, they conducted the harmonic analysis to demonstrate that the PCB vibration is reduced by placing the capacitors in the least sensitive region.

The same authors propose a statistical simulation method and a parameter sensitivity analysis in another paper [35]. The statistical simulation determines the PCB intrinsic vibration properties, including the parameter variation effect [36]. The parameter sensitivity analysis is performed to understand the dominant parameters controlling the PCB vibration properties. They consider the tolerances given by the PCB suppliers for the board material properties (mass density, Young’s modulus, and Poisson’s ratio), together with the PCB dimensions, to obtain the natural frequencies and the corresponding modal shapes. The MLCCs effect on the board vibration is investigated through harmonic analysis applying modal superposition. The simulated natural frequencies of the PCB and the influence of the MLCCs vibration on the board are contra-validated through measurements, indicating the capability of the proposed simulation methods.

Wand et al. [6] chose to construct a three-dimensional finite element model (FEM) of an MLCC. They neglected the capacitor fillets due to the insignificant influence on the MLCC vibration and, as hundreds of dielectric layers are present in the capacitor structure, they simplified the model with fewer layers but similar vibration performance. They ignored the electrostriction effect and simplified Equation (1) into Equation (9) to express the mechanical strain on *z* direction *s_z_*, and into Equation (10) to represent the mechanical strain on *y*-direction *s_y_*. The *x* direction was neglected due to its similarity with the *y* direction.
(9)$$s_{z}=d_{33}\cdot E=\frac{\Delta H}{H},$$
(10)$$s_{y}=d_{31}\cdot E=\frac{\Delta L}{L},$$
where *d_33_* and *d_31_* are the piezoelectric coefficients on *z* and *y* direction, respectively, *H* and *L* are the thickness and the length of a single dielectric layer, and the *z* direction and *y* direction deformations of a single dielectric layer are represented by ∆*H* and ∆*L*, respectively.

The deformation of a single dielectric layer can be calculated depending on the electrical load *V*, as presented in Equations (11) and (12):(11)$$\Delta H=d_{33}\cdot V,$$
(12)$$\Delta L=\frac{L}{H}\cdot d_{31}\cdot V.$$

When all the *N* MLCC layers are taken into consideration, the deformations can be approximated as Equations (13) and (14):(13)ΔH=H·d33·VHN=d33·V·N,
(14)$$\Delta L=\frac{L}{H}\cdot d_{31}\cdot V\cdot N.$$

For the simulation simplification, the authors consider a reduced number of dielectric layers *N’*, while the displacement ∆*H* remains invariant. Therefore, to maintain the vibration performance of the MLCC, they changed the piezoelectric coefficient using the correlations represented in Equations (15) and (16):(15)$$\Delta H=d_{33}\cdot V=d_{33}^{\prime }\cdot V\cdot N^{\prime },$$
(16)$$d_{33}^{\prime }=\frac{N}{N^{\prime }}\cdot d_{33}.$$

The MLCC model was validated with a vibration test, and the results had an error of under 7%.

Based on vibration analysis, the authors studied the solder joint effect on the PCB vibration caused by MLCCs. Their theory is that the singing capacitors phenomenon can be attenuated by eliminating the links between the MLCC termination and PCB, as presented in Figure 13a,b.

The authors of Ref. [6] demonstrated, both by simulation and experiment, that this soldering method improves the vibration characteristics transmitted from the MLCC to the PCB. For the experimental validation, the soldering was performed manually. They placed a plastic sheet between the capacitor and PCB. Afterward, the plastic sheet was removed. Unfortunately, this process is hard to implement in mass production manufacturing.

## 5. Discussion

Due to the piezoelectric and electrostrictive effects of BaTiO_3_, the MLCCs’ inter electrodes vibrate, causing a chain reaction. The vibration is transferred from the inter electrodes to the capacitor terminals, from the terminals to the solder joint, and finally from the solder joint to the PCB, causing the singing capacitor phenomenon. Therefore, to eliminate the audible noise caused by MLCC, we must interrupt the vibration transfer. 

The first solution would be to use capacitors with a low dielectric constant. These capacitors would solve the singing capacitor phenomenon from the root cause.

For the vibration transfer from MLCC to the solder joint, many alternatives are available. The component suppliers offer low acoustic noise capacitors, such as metal terminals or metal plate capacitors, interposer or alumina substrate MLCCs, thicker dielectric layer capacitors, and dipped radial leads capacitors. The most effective commercial solution is the metal terminal capacitor, which attenuates the noise by 25 dB. Some authors suggest increasing the cover layer thickness to reduce the transmitted vibration.

For the vibration transfer to the PCB, some authors suggested decreasing the solder joint. Unfortunately, to date, this has not been demonstrated to be efficient. Other authors suggested placing the MLCC in a vertical orientation on PCB to increase the resonating frequency. Of course, the layout geometry has a big impact on the PCB acoustic noise. The mirror or back-to-back configuration is considered the best solution in the literature.

As mentioned before, it is better to prevent than to correct. By simulating the design, we can avoid the apparition of the singing capacitor phenomenon. To simulate the acoustic behavior, we need a harmonic analysis. The simulation is more precise if we also implement a statistical simulation and a parameter sensitivity analysis. Some authors recommend a three-dimensional FEM simulation and a vibration analysis [37].

The two most popular methods found in the literature to measure the acoustic noise caused by MLCCs are SPL measurement and LDV measurement. These two can be correlated with a linear equation. Other methods to investigate the singing capacitor phenomenon are the optical sensor fiber, piezoelectric accelerometer, active excitation method, and vibration and rail voltage coherence method.

In conclusion, the acoustic noise caused by MLCCs is a current problem in modern electronic systems. Therefore, the interest in this issue is high among the experts who study electronics in specialized literature. Unfortunately, the solution for this phenomenon is not straightforward due to the design-oriented behavior of the MLCCs.

## Figures and Tables

**Figure 1 sensors-22-03869-f001:**
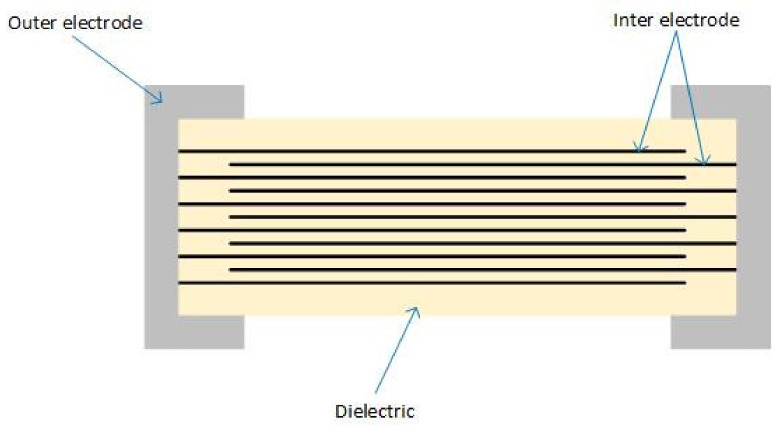
MLCC Structure.

**Figure 2 sensors-22-03869-f002:**
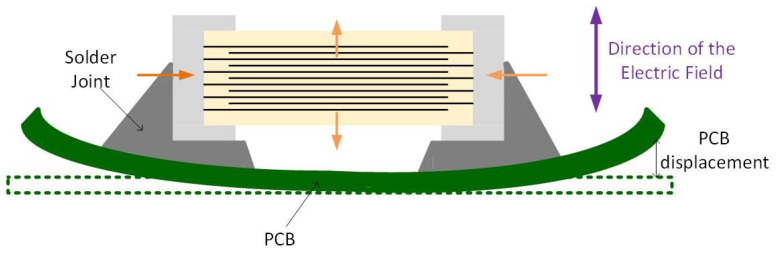
The mechanism for singing capacitor phenomenon.

**Figure 3 sensors-22-03869-f003:**
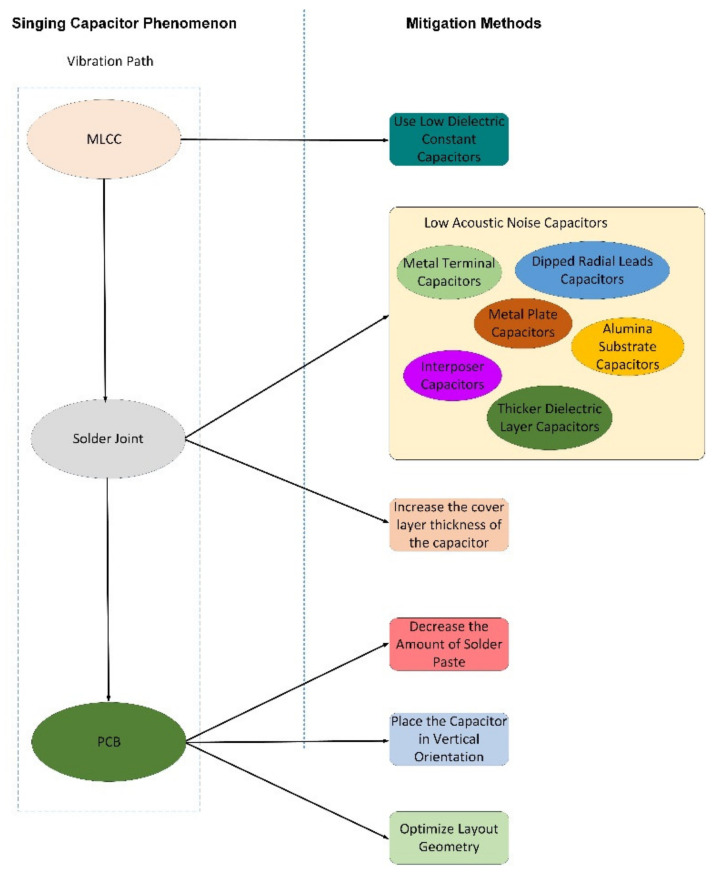
Solutions for singing capacitor phenomenon.

**Figure 4 sensors-22-03869-f004:**
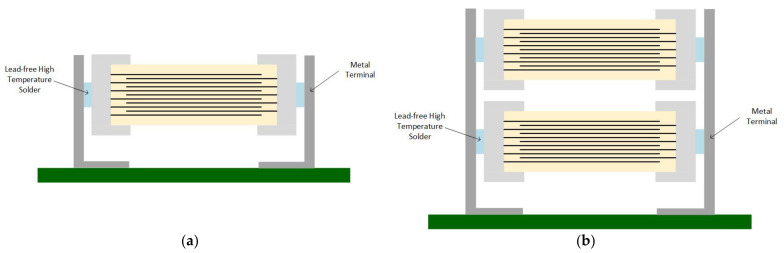
Metal terminal capacitors (**a**) Single MLCC type; (**b**) Dual MLCC Type.

**Figure 5 sensors-22-03869-f005:**
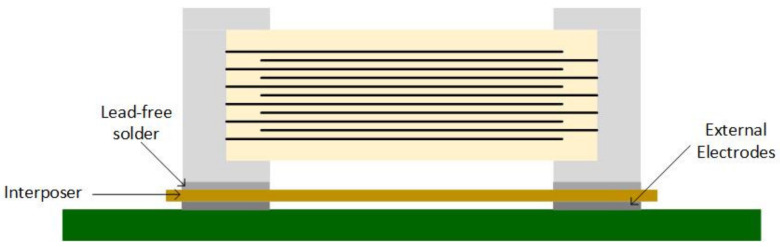
Interposer Capacitor.

**Figure 6 sensors-22-03869-f006:**
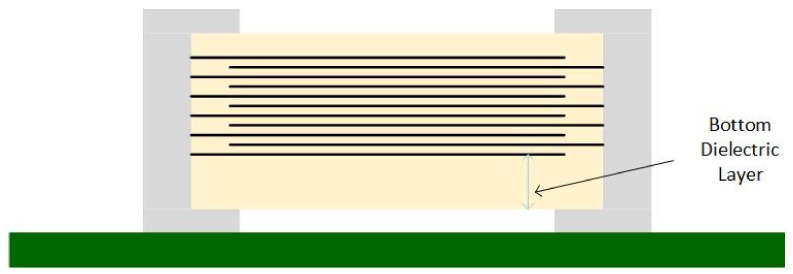
Thicker Bottom Dielectric Capacitor.

**Figure 7 sensors-22-03869-f007:**
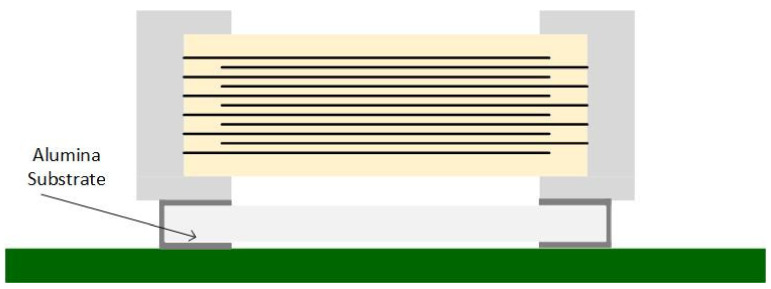
Alumin Substrate Capacitor.

**Figure 8 sensors-22-03869-f008:**
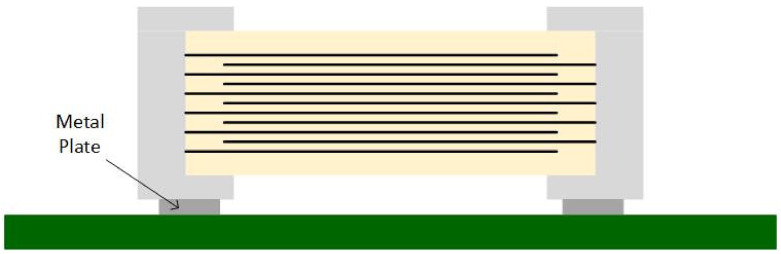
Metal Plate Capacitor.

**Figure 9 sensors-22-03869-f009:**
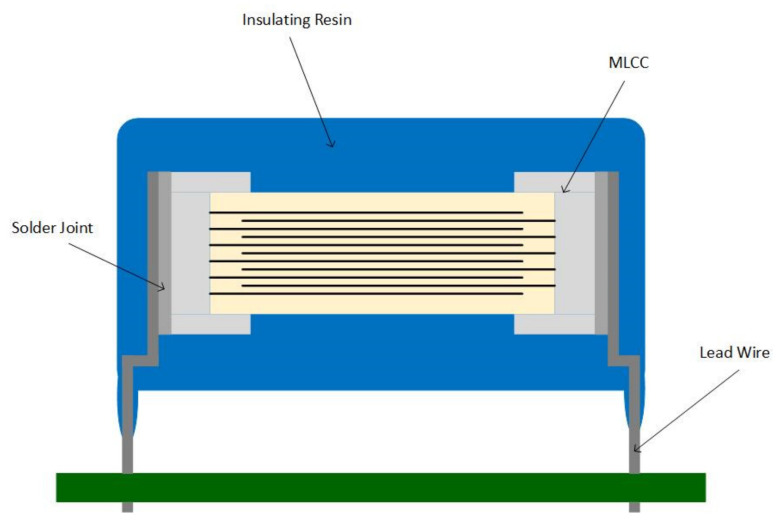
Dipped Radial Leads Capacitor.

**Figure 10 sensors-22-03869-f010:**
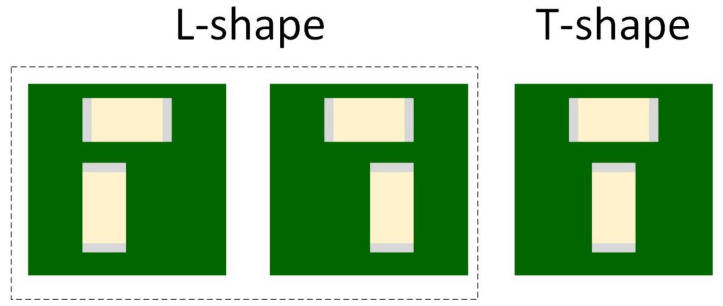
L-shape and T-shape layout geometry.

**Figure 11 sensors-22-03869-f011:**
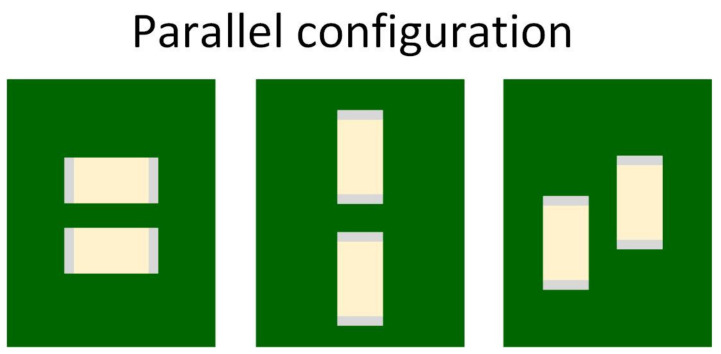
Parallel layout geomery.

**Figure 12 sensors-22-03869-f012:**
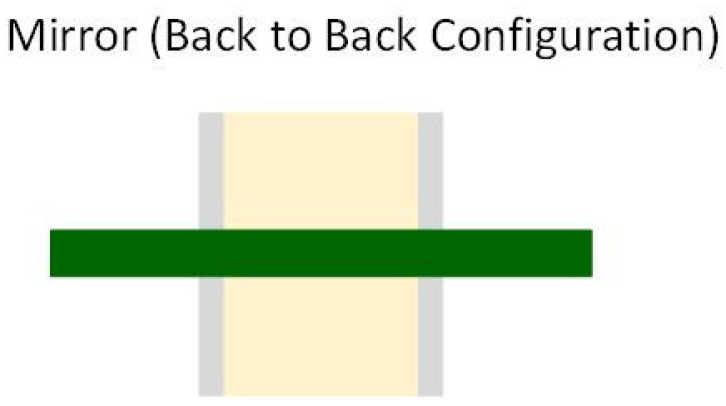
Mirror or Back-to-Back layout geometry.

**Figure 13 sensors-22-03869-f013:**
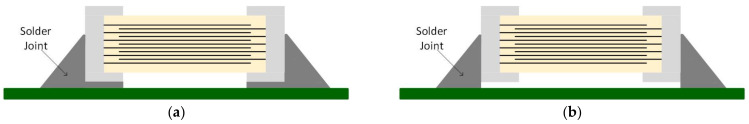
MLCC soldering (**a**) Classical soldering; (**b**) Proposed soldering.

**Table 1 sensors-22-03869-t001:** Methods to reduce the acoustic noise created by MLCCs.

Layout Geometry	Regular		3-Terminal		Reverse Geometry		Interposer	
	In Phase	Out of Phase	In Phase	Out of Phase	In Phase	Out of Phase	In Phase	Out of Phase
**L-shape/T-shape**	NO	NO	YES	NO	NO	NO	NO	YES
**Parallel**	NO	YES	NO	YES	NO	YES	NO	YES
**Back-to-Back**	YES	NO	NO	NO	YES	NO	YES	NO

The table contains the answer to the question “Is this combination effective?”.

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
