# Peer review of "“Singing” Multilayer Ceramic Capacitors and Mitigation Methods—A Review"

_sensors, 2022, doi:10.3390/s22103869_

Round 1

Reviewer 1 Report

This manuscript presents a comprehensive literature review of the acoustic noise caused by MLCCs in electronic devices, containing measurements methodologies, solutions and simulation methods. Additionally, the advantages and disadvantages of different structures are compared. The results of this paper are comprehensive and have potential reference values that deserve to be widely circulated. I am glad to recommend it for publication. However, some revisions are needed.

  1. Among several structural designs, which structural design is the most effective for weakening the singing capacitors phenomenon, please discuss clearly.
  2. It is mentioned in the article that solder paste can weaken the acoustic noise caused by MLCCS. Is there any other material that may be better than solder paste? it is suggested to increase the discussions.

Reviewer 2 Report

The organization and writing are poor and unprofessional. The topic is a good one for a review but the length of the paper can be half of what it is and still provide the review. I have some examples and suggestions below.

The last sentence of the paper makes no sense: "This issue is studied by many experts in the literature, and the solutions to prevent, measure, and resolve this phenomenon are design-oriented described." 

Corrections will be needed for some chemical formulas with appropriate subscripts - such as BaTiO3

It is wrong to state (line 81) that "the induced vibration is transferred to the PCB via solder paste [1,16]" - solder paste is not present after assembly. It is a solder joint.

Do not use single-sentence paragraphs as in lines 100-101.

Line 88 - "three major guilty factors" - guilty is too colloquial a word.

In many places, the writing and composition is immature and unprofessional. Example: "Although this method could be used to measure the PCB vibration caused by the MLCCs, the amount of information available in the literature is little. Moreover, the existing papers are outdated. For example, Perrone and Vallan [17] suggest that LDV is not precise enough to measure very small displacements, while Sun et al. [9,10] demonstrate the opposite."

Most of the basic physical equations are not necessary for the scope of this paper. I think that the equations 1-11 and accompanying discussions can be deleted without the loss of any clarity.

In the solutions section, the company-provided information is presented without critical review and it reads like an advertisement. Also, a discussion like TI warns about costs are trivial and do not belong to a paper. The whole section 3.1 should be based on the method of noise elimination rather than a listing of products from companies with unnecessary sketches.

Figure 13 is a good summary and it will be better to organize the paper with that image as a guide. Those bubble texts can be the section titles.

Reviewer 3 Report

In this work, the authors conclude the vibration phenomenon of MLCC device in practical application and reviewed some mitigation mecthods. It is an interesting and novel topic. Some issues should be addressed before this manuscript being considered for publication.

1. In general, the strength of piezoelectric vibation reaches the maximum at resonance frequencies (Ref. J. Appl. Phys., 112, 106102, 2012; ,Adv. Mater. 20, 4776, 2008). This work does not mention this point. A deep dicsussion on the singing of MLCC at resonance frequencies should be given.

2. The principle and physical mechanism of the mitigation mecthods such as L-shape, T-Shape, Parallel or Back-to back geometry should be analyzed and clearly given (Ref. Ceram. Int., 47, 17147, 2021).

3. The resonance frequency of piezoelectric vibration is related to the size. Are these mitigation mecthods still effective for the MLCC devices with different sizes? 

4. It's better to also discuss the influence of temperature on the singing of MLCC  (Ceram. Int., 46, 27579, 2020; J. Adv. Cearam. 9,584, 2022 ).

Round 2

Reviewer 2 Report

Except for moving the final picture to a different location, I do not see any of my review comments being implemented. Even the basic wrong usage of "Solder Paste" instead of "Solder Joint" is not fixed.

Author Response

We are very sorry for not correcting solder paste with solder joint in our paper. Due to a misunderstanding, we corrected the mistake only in the location suggested in the review report. Now you should find the correct variant in our paper in all appropriate positions.

We have addressed your comments in our manuscript, as stated in both our review response and cover letter. We have also removed equations 7 - 9 and the accompanying discussions in this revision. The only points not 100% implemented now are 7 (we did not delete all equations) and 8 where we still believe the limited reference to commercial solution is necessary for the reader, as explained in our previous answer.

We hope that the current manuscript reaches your expectations.

Reviewer 3 Report

The revised manuscript is ok now.

Author Response

Thank you very much for your kind comment.